# Regulatory Role of Orexin in the Antistress Effect of “Press Tack Needle” Acupuncture Treatment

**DOI:** 10.3390/healthcare9050503

**Published:** 2021-04-27

**Authors:** Aki Fujiwara, Mana Tsukada, Hideshi Ikemoto, Takuji Izuno, Satoshi Hattori, Takayuki Okumo, Tadashi Hisamitsu, Masataka Sunagawa

**Affiliations:** 1Department of Physiology, School of Medicine, Showa University, Tokyo 142-8555, Japan; fujiwara@tenshinotamago.com (A.F.); m-tsukada@med.showa-u.ac.jp (M.T.); h_ikemoto@med.showa-u.ac.jp (H.I.); tizunosuke19831113@gmail.com (T.I.); bin7770@me.com (S.H.); tokumo@med.showa-u.ac.jp (T.O.); tadashi@med.showa-u.ac.jp (T.H.); 2Acupuncture & Moxibustion Clinic Tenshinotamago, Tokyo 104-0061, Japan

**Keywords:** orexin, acupuncture, press tack needle, antistress effect, orexin receptor, aggressive behavior

## Abstract

The aim of this research was to investigate the antistress effect of press tack needle (PTN) acupuncture treatment using rats with social isolation stress (SIS). Rats were divided into non-stress group (Grouped+sham), stress group (SIS+sham), and PTN-treated SIS group (SIS+PTN). Rats in the SIS+PTN and SIS+sham groups were housed alone for eight days. For the SIS+PTN group, a PTN (length, 0.3 or 1.2 mm) was fixed on the GV20 acupoint on day 7. We measured stress behavior based on the time the rats showed aggressive behavior and the levels of plasma corticosterone and orexin A on day 8. In addition, the orexin-1 receptor or orexin-2 receptor antagonist was administered to rats that were exposed to SIS. The duration of aggressive behavior was significantly prolonged in the SIS+sham group, and the prolonged duration was inhibited in the SIS+PTN (1.2 mm) group. The levels of plasma corticosterone and orexin A were significantly increased in the SIS+sham group; however, these increases were inhibited in the SIS+PTN group. The aggressive behavior was significantly reduced after the orexin-2 receptor antagonist was administered. These findings suggest that PTN treatment at GV20 may have an antistress effect, and the control of orexin is a mechanism underlying this phenomenon.

## 1. Introduction

Stressors cause stress responses via the hypothalamic–pituitary–adrenal (HPA) and sympathetic–adrenal–medullary (SAM) axes [1,2]. The stress response is known as the “fight-or-flight” response, and it was originally a necessary reaction for survival. However, it is accompanied with negative emotions, such as anxiety, nervousness, and anger, and repeated and/or long-term exposure to stressful situations may be related to the development of psychological disorders, such as depression and neurosis. To prevent this, it is important to control negative emotions and minimize stress in daily life. One of the means is the regulation of orexin secretion.

Orexins (also called hypocretins) are hypothalamic neuropeptides, and orexins A and B are derived from the common precursor peptide, prepro-orexin [3,4]. Their neurons mainly are located in the lateral and perifornical areas of the posterior hypothalamus. Their axons widely project throughout the central nervous system, except in the cerebellum [3]. Orexins bind to the orphan G-protein coupled receptors orexin-1 receptor (OX1R), which is highly selective for orexin A, and orexin-2 receptor (OX2R), which is nonselective for orexin A or B [4]. Orexins regulate multiple physiological functions such as arousal [5,6], sleep [5,6], feeding [7], metabolism [8], analgesia [9,10] and the autonomic regulation of the cardiovascular [11,12], respiratory [13] and neuroendocrine systems [14,15]. Orexin was recently reported to be involved in the regulation of the HPA and SAM axes [16,17]. Previous studies that used rodent models of stress revealed that immobilization stress, cold stress [18], swimming stress [19], or social isolation stress (SIS) [20] activate the orexin neuron. Intraventricular administration of orexin increases the plasma levels of the adrenocorticotropic hormone (ACTH) and corticosterone [21], and the systemic administrations of both orexins directly act on adrenocortical cells and increases the plasma levels of aldosterone and corticosterone [22]. Moreover, orexin receptors are expressed in the adrenal medulla, and orexin stimulates adrenaline release [23]. These findings suggest that orexin plays a crucial role in controlling the stress response.

Acupuncture is used for various diseases and symptoms and has been adapted to treat psychiatric disorders, including anxiety disorders, depression, bipolar disorder, schizophrenia, drug dependence, eating disorders, and sleep disorders caused by stress [24,25]. Several animal studies reported that acupuncture treatments control stress responses and activate the HPA and the SAM axes in response to varying degrees of stress (Table 1) [26,27,28,29,30,31]. The use of acupuncture and methods of acupuncture stimulation vary depending on the types of stress; however, excessive activations of the HPA and SAM axes can be regulated.

The press tack needle (PTN; also called the thumb tack needle) is a short acupuncture needle approximately 0.3–1.5 mm in length. PTN is fixed onto the skin with a bandage plaster, and its effectiveness against muscle pain, muscle fatigue, menstrual pain, sleep disorders, and body complaints of dialysis patients has been well reported [32,33,34,35,36,37]. Because PTN is short, safe, and easy to use, it can also be used for self-care. There are few reports on the antistress effect of PTN treatment [37], and the mechanism of action has not been clarified. Moreover, the difference in actions depending on lengths has not been clarified by clinical or basic research, and the length of PTN used was chosen according to the practitioner’s experience.

Therefore, we investigated the antistress effects of PTN treatment and the difference in those effects depending on the length of PTN using an animal model of chronic psychological stress. We exposed rats to SIS following the treatment of PTN and examined whether PTN could have inhibitory effects on SIS-induced aggressive behavior [20]. Furthermore, we investigated whether orexins contributed to the mechanism underlying the effect.

## 2. Materials and Methods

### 2.1. Animals

Male Wistar rats (7 weeks old, weighing 180–200 g) were purchased from Nippon Bio-Supp. Center (Tokyo, Japan). During the experiment, animals were housed in standard plastic cages (for a group habitat, W 24 cm × L 40 cm × H 20 cm; and for a single habitat, W 26 cm × L 26 cm × H 18 cm) and were kept in our animal facility at 25 °C ± 2 °C, 55% ± 5% humidity, with a light/dark cycle of 12 h. Food (CLEA Japan, CE-2, Tokyo, Japan) and water were provided ad libitum. The experiments were performed according to the guidelines of the Committee of Animal Care and Welfare of Showa University. All experimental procedures were approved by the Committee of Animal Care and Welfare of Showa University (certificate number: 07063; date of approval: 1 April 2017). All efforts were made to minimize animal suffering and to use the minimum number of animals necessary to conduct this study with reliable results.

### 2.2. Drugs

SB334867, an orexin-1 receptor antagonist (OX1RA) (Tocris Bioscience, 792173-99-0, Bristol, UK), and TCS OX2 29, an orexin-2 receptor antagonist (OX2RA) (Tocris Bioscience, 372523-75-6), were dissolved in 10% dimethyl sulfoxide (DMSO) (Wako Pure Chemical Industries, 043-07216, Osaka, Japan), 10% 2-hydroxypropyl-β-cyclodextrin (HCD) (Sigma-Aldrich, C0926, St. Louis, MO, USA) and 80% distilled water, and intraperitoneally administered [38,39,40].

### 2.3. PTN

Figure 1A shows the structure of the PTN used in this study (Seirin Corporation, Pyonex, Shizuoka, Japan). Each PTN involved a needle with a diameter of 0.11 mm and length of 0.3 mm (PTN0.3), and diameter of 0.20 mm and length of 1.2 mm (PTN1.2). A sham PTN, whose needle element was removed, was used as the placebo.

### 2.4. Influence of the PTN Treatment on the Plasma Orexin A Level

The plasma orexin A level was used as an indicator of the influence of the PTN treatment on orexin secretion. Eighteen rats (group-housed without stress load) were randomly divided into three groups (each *n* = 6): Grouped+sham, Grouped+PTN0.3, and Grouped+PTN1.2. Every rat was lightly anesthetized using intraperitoneally administered pentobarbital sodium (30 mg/kg; Somnopentyl, Kyoritsu Seiyaku Co., Tokyo, Japan). The hair of the head was shaved using hair clippers. A PTN or a sham PTN was fixed on the acupuncture point that corresponds to the human Governor Vessel 20 (GV20; Baihui) acupoint, which is located above the apex auriculae on the midline of the head (Figure 1B). A PTN of each length was attached to the rats in the PTN group, and a sham PTN was attached to rats in the Grouped+sham group. On the following day, rats were anesthetized using intraperitoneally administered pentobarbital sodium (50 mg/kg), and blood samples were taken from the inferior vena cava. To avoid influences of fluctuation in routine, PTN fixing and blood sampling were performed from 1:00 to 3:00 p.m. The plasma was separated by centrifugation and was immediately frozen at −80 °C and stored until analysis. The plasma orexin A level was measured using an enzyme-linked immunosorbent assay (ELISA) kit (EKE-003-30, Phoenix Pharmaceuticals, Burlingame, CA, USA) according to the manufacturer’s instructions.

### 2.5. Antistress Effect of the PTN Treatment

#### 2.5.1. Groups and Stress Procedure

In the present study, a rat model of SIS was used. This animal model was used to study psychological stress that is induced by a loss of social interaction. Wild rats live in groups, and when housed alone, they show stress responses, such as adrenomegaly; increased secretions of ACTH, corticosterone, and catecholamine [41,42]; aggressive behavior [20,43]; locomotor hyperactivity [43,44]; increased food intake [45]; and oxidative stress [46]. In the present study, we evaluated aggressive behavior and corticosterone secretion. Twenty-four rats were randomly divided into groups of four as follows: group-housed (Grouped+sham), individually housed stress (SIS+sham), PTN0.3-treated SIS (SIS+PTN0.3), and PTN1.2-treated SIS (SIS+PTN1.2) groups. Rats in the SIS+sham and SIS+PTN groups were individually housed for 8 days, and those in the Grouped+sham group were housed in groups of two or three per cage. On day 7, a PTN or sham PTN was fixed on the GV 20 acupuncture point following the same methods described above. Rats in the SIS+PTN groups were attached with a PTN of each length, and those in the Grouped+sham and SIS+sham groups were attached with the sham PTN. The protocol is shown in Figure 2.

#### 2.5.2. Aggressive Behavior Test

On day 8, the anti-aggressive effects of the PTN treatment were investigated in SIS-exposed rats as an evaluation of the antistress effect [20]. Male Wistar rats (5 weeks old, weighing 150–160 g) were housed in a group of three and were used as intruders. An intruder rat was placed in a resident’s home cage, and resident–intruder interactions were recorded using a video camera for 10 min. Resident rats subjected to SIS showed aggressive behavior characterized by wrestling, boxing, and biting. The behaviors of tail rattling, lateral threat, and pursuit were excluded. The time spent exhibiting aggressive behavior during a 10 min observation period was measured. This test was performed from 11:00 a.m. to 1:00 p.m.

#### 2.5.3. Plasma Orexin A and Corticosterone Levels

Levels of plasma orexin A were measured to record its involvement in the stress response phenomenon, and levels of plasma corticosterone were measured to evaluate psychological stress. On day 8, 2 h after the aggressive behavior test, blood sampling was performed (from 1:00 to 3:00 p.m.) using the same methods as described above, and plasma levels of orexin A and corticosterone were measured using ELISA kits (orexin A, EKE-003-30, Phoenix Pharmaceuticals; corticosterone, ADI-900-097, Enzo Life Sciences, Farmingdale, NY, USA) according to the manufacturer’s instructions.

#### 2.5.4. Immunofluorescence of Orexin A

Orexin secretion in the lateral hypothalamus was determined using immunofluorescence analysis. After blood sampling, rats were exsanguinated by intracardial perfusion of phosphate-buffered saline (PBS) at pH 7.4. Rat brains were harvested after perfusion with 4% paraformaldehyde in 0.1 M PBS. Tissue specimens were embedded in an O.C.T. compound (Sakura Finetek, Tissue-Tek, Tokyo, Japan), frozen, and then cut into 20-µm-thick sections using a cryostat (Leica Biosystems, CM3050S, Nussloch, Germany). Sections were incubated overnight at 4 °C with a rabbit anti-orexin A antibody diluted to 1:100 (Merck, PC362, Anti-Orexin A Rabbit Antibody, Merck, Darmstadt, Germany) and then incubated with Alexa Fluor 555-conjugated donkey anti-rabbit IgG secondary antibody diluted to 1:1000 (Thermo Fisher Scientific, Waltham, MA, USA) for 2 h at room temperature. Nuclei were counterstained with 4′,6-diamidino-2-phenylindole (DAPI, 1:1000; Thermo Fisher Scientific). Samples were imaged using a confocal laser scanning fluorescence microscope (FV1000D, Olympus, Tokyo, Japan). For each rat, 10 orexinergic neurons were randomly selected from the lateral area of the posterior hypothalamus per rat, and the optical densities of immunoreactive staining were measured using an appropriate software program (FV10-AW, Olympus). An average of 10 neurons was used as the individual value and compared. The background immunofluorescence was measured in the absence of a primary antibody.

### 2.6. Involvement of the Orexin Receptor in Aggressive Behavior

To determine whether orexin receptors are involved in aggressive behavior, rats were randomly divided into groups of four as follows: group-housed control group (Grouped; *n* = 4), individually housed stress group (SIS; *n* = 4), OX1RA-treated SIS group (SIS+OX1RA; *n* = 4), and OX2RA-treated SIS group (SIS+OX2RA; *n* = 4). Rats in the SIS groups were individually housed for seven days, and those in the Grouped group were housed in groups of three per cage. On day 7, rats in the SIS+OX1RA group were intraperitoneally administered with SB334867 (20 mg/kg) [38,47] 30 min before the test, and those in the SIS+OX2RA group were administered with TCS OX2 29 (20 mg/kg) [47] 30 min before the test. Rats in the other groups were administered with only the vehicle (DMSO, HCD and distilled water). An aggressive behavior test was performed as described above.

### 2.7. Statistical Analysis

All experimental data were presented as mean ± standard deviation. The statistical significance of the differences among groups was evaluated using one-way analysis of variance (ANOVA). Post hoc comparisons between the groups were performed by Tukey’s test or the Tukey–Kramer test using SPSS 25 (IBM Japan, Tokyo, Japan). Correlations were analyzed with Spearman’s rank correlation coefficient. All *p*-values of <0.05 were considered statistically significant.

## 3. Results

### 3.1. Influence of PTN on the Plasma Orexin A Level

Influences of PTNs (lengths, 0.3, and 1.2 mm) on orexin A secretion were investigated in healthy rats without stress. The plasma orexin A levels showed a decreasing tendency in accordance with their lengths; however, there were no significant differences among the groups (*F*(2, 15) = 2.83, *p* = 0.09; *p* = 0.53, Grouped+sham vs. Grouped+PTN0.3; *p* = 0.08, Grouped+sham vs. Grouped+PTN1.2) (Figure 3).

### 3.2. A Level Aggressive Behavior Test

On day 8, we measured the total duration of aggressive behavior during a 10 min observation. Rats in the SIS+sham group exhibited significantly longer durations of aggressive behavior compared with rats of the Grouped+sham group (*F*(3, 20) = 8.64, *p* < 0.001; Grouped+sham, 17.83 ± 31.99 s; SIS+sham, 203.33 ± 83.11 s, *p* < 0.01). However, the PTN treatment (1.2 mm) significantly inhibited aggressive behavior (92.50 ± 53.71 s, *p* < 0.05) (Figure 4A). The PTN treatment (0.3 mm) also showed the inhibitory tendency, but there was no significant difference (114.50 ± 73.42 s, *p* = 0.11).

### 3.3. Plasma Level and Immunofluorescent Staining of Orexin A

The plasma orexin A level significantly increased in the SIS+sham group compared with that of the Grouped+sham group (*F*(3, 20) = 10.23, *p* < 0.001; Grouped+sham, 0.23 ± 0.05 ng/mL; SIS+sham, 0.45 ± 0.10 ng/mL, *p* < 0.01). However, PTN treatment (1.2 mm) significantly inhibited the elevation of orexin A level in response to stress (0.21 ± 0.05 ng/mL, *p* < 0.01; vs. SIS+sham group) (Figure 4B). PTN treatment (0.3 mm) did not show a significant inhibitory effect (0.33 ± 0.12 ng/mL, *p* = 0.10; vs. SIS+sham group).

Immunohistochemical analysis revealed increased orexin A production in the lateral hypothalami of rats in the SIS+sham group compared with those of the Grouped+sham group; however, this increase was inhibited in the SIS+PTN1.2 group. Representative pictures are shown in Figure 4C. The values were then expressed as optical densities (Figure 4D) (*F*(3, 20) = 43.04, *p* < 0.001). The level was significantly increased in the SIS+sham group (1326.53 ± 312.44 IR Density) compared with that in the Grouped+sham group (115.17 ± 54.17 IR Density) (*p* < 0.01). This increase was significantly inhibited by the PTN treatment (PTN0.3, 635.14 ± 277.04 IR Density; PTN1.2, 153.53 ± 20.69 IR Density; *p* < 0.01). Considered together, these findings indicate that the PTN treatment attenuates the promotion of orexin secretion caused by social isolation stress.

### 3.4. Consideration of the Correlation between Attack Duration and Orexin A Level

The correlation between attack duration and orexin A level was investigated. A significantly positive linear correlation was observed between them (*ρ* = 0.57, *p* < 0.01) (Figure 4E), which indicated that orexin A induces aggressive behavior.

### 3.5. Plasma Corticosterone Level

The plasma corticosterone level, which was measured as a marker of stress, significantly increased in the SIS+sham group compared with that in the Grouped+sham group (*F*(3, 20) = 9.12, *p* < 0.001; Grouped+sham, 46.05 ± 17.66 ng/mL; SIS+sham, 158.50 ± 59.29 ng/mL, *p* < 0.01). However, the PTN treatments significantly inhibited the elevation of corticosterone level in response to stress (SIS+PTN0.3, 79.43 ± 51.98 ng/mL, *p* < 0.05; SIS+PTN1.2, 52.41 ± 22.34 ng/mL, *p* < 0.01) (Figure 5).

### 3.6. Involvement of Orexin Receptor in Aggressive Behavior

To determine whether orexin receptors are linked to aggressive behavior, we measured the total duration of aggressive behavior following the administration of OX1RA or OX2RA. Rats in the SIS group showed significantly longer durations of aggressive behavior compared with that of rats in the Grouped group (*F*(3, 12) = 38.99, *p* < 0.001; Grouped, 0.00 ± 0.00 s; SIS, 160.73 ± 33.31 s, *p* < 0.01); however, the OX2RA treatment significantly inhibited aggressive behavior (39.04 ± 19.10 s, *p* < 0.01 vs. SIS and SIS+OX1RA) (Figure 6). The OX1RA treatment also showed an inhibitory effect, although there was no significant difference (117.93 ± 26.80 s, *p* = 0.10).

## 4. Discussion

In the present study, we investigated the antistress effect of the PTN treatment at GV20 to identify whether the length of PTN resulted in significant differences in the indicators used to evaluate antistress activities. Orexin was an important focus as a potential mechanism underlying the antistress effect. The 1.2-mm PTN treatment at GV20 significantly inhibited aggressive behaviors induced by SIS (Figure 4A), and the 0.3-mm PTN treatment also showed an inhibitory tendency (*p* = 0.11). The PTN is a popular acupuncture needle, because it does not require special training and is relatively safe as well as easy to use. However, there have been no reports that compare the differences in effect depending on the length of the PTN. The length is usually determined by the practitioner’s experience. The results of this study revealed that a higher amount of stimulation was correlated with better results, which were contingent on whether the patient complained of pain or unpleasant symptoms. Further studies are needed to determine the optimal PTN length.

GV20 is a prime acupuncture point that belongs to the Governor Vessel meridian and is clinically used to treat psychological disorders, depression, headache, stiff shoulder, ear noise, dizziness, and nasal congestion [48]. In clinical practice, acupuncture points are used according to the patient’s condition. In a preliminary experiment, we applied PTN at the bladder 21 (BL21, Weishu) acupuncture point, which is located on the upper back region, to confirm if another acupoint provides a similar antistress effect [49]; however, this was not evident (data not shown). Tanahashi et al. [31] reported that depression-like behaviors induced by the water-immersion stress in rats improved, and the serum corticosterone level decreased following acupuncture stimulation at the GV20 and the extraordinary points of the head and neck (Ex-HN3, Yintang). The stimulus was continued for 20 min a day with the PTN inserted to a depth of 5 mm using stainless-steel needles (diameter, 0.25 mm; length, 15 mm). Zhang et al. [50] reported that the acupuncture treatment was carried out for 10 min a day at GV20, and Ex-HN3 ameliorated depression-related behavior and promoted neurogenesis in the hippocampus of rats exposed to chronic unpredictable mild stress. In this study, stainless-steel needles of 0.3 mm diameter were inserted 2–3 mm deep. Our results indicated that mild stimulation at GV20 is still effective; therefore, shallow stimulation is effective for relieving stress. However, our treatment was administered at longer durations during the day, and the types of stressors were different. Thus, further investigation in other models is needed.

Many clinical studies have reported that acupuncture treatment has an antistress effect and controls the activation of the HPA and SAM axes [26,27,28,29,30,31,51]; however, the biological mechanism was not clarified. As mentioned above, orexin has been reported to be involved in the regulation of the HPA and SAM axes [16,17,18,19,20,21,22,23]. As described above, orexins A and B are derived from the same precursor peptide, prepro-orexin [3,4], and orexin A shows similar affinities to OX1R and OX2R, whereas orexin B mainly shows affinity to OX2R [4]. We previously confirmed that, in the same model rat, both orexins exhibited the same secretory tendencies [20]. In the present study, we investigated changes in the secretion of orexin A. In healthy rats, the PTN treatment did not show a significant change in orexin A secretion (Figure 3); however, interestingly, in rats with mental stress, enhanced secretion of orexin A was significantly suppressed by the 1.2 mm PTN treatment (Figure 4B,C).

There are some reports about the influences of acupuncture on orexin secretion [52,53,54,55]. Orexin A secretion and mRNA expressions of OX1R and OX2R in lung tissue increased in chronic obstructive pulmonary disease model rats; however, these increases were suppressed by electroacupuncture treatment (10/50 Hz) at BL13 (Feishu) and Stomach 36 (ST36, Zusanli) [52]. Furthermore, the activation of orexin neurons by the morphine administered to rats was suppressed by electroacupuncture treatment (2 Hz) at ST36 and BL23 (Shenshu), and the rewarding effect of morphine was attenuated [53]. On the other hand, a decrease in orexin A secretion in rats with acute pain (laparotomy pain) was inhibited by electroacupuncture treatment (2/15 Hz) at ST36 and Spleen 6 (SP6, San Yin Jiao), and pain was reduced [54]. In addition, electroacupuncture treatment (2 Hz) at Pericardium 6 (PC6, Neiguan) in mice promoted orexin A secretion and suppressed thermal hyperalgesia and mechanical allodynia due to neuropathy [55]. Acupuncture has different effects on orexin secretion depending on the pathological condition, although the acupuncture points used and the methods of stimulation are different. Further research is needed to elucidate this mechanism.

A significant correlation was detected between the plasma orexin A concentration and attack duration (Figure 4E). However, there is one limitation in that the orexin A concentration was measured 2 h after the aggressive behavior test. We previously measured orexin levels in the same animal model on a different day after the aggressive behavior tests [20]. Then, the orexin level had significantly increased due to isolated stress. Therefore, the orexin secretion increased regardless of the behavior test. In other words, orexin secretion was increased by isolated stress loading; as a result, aggressive behavior was induced. To clarify this, an aggressive behavior test was conducted after an orexin receptor antagonist was administered following SIS. As a result, OX2RA administration significantly suppressed the aggressive behavior (Figure 4). OX2RA was reported to be related to the regulation of intermale aggressive behavior [56] and maternal aggression caused by social stress during lactation [57]. These findings suggest that SIS enhanced orexin secretion and excitability, especially via OX2R; however, PTN treatment suppressed orexin secretion and aggressive behaviors. In the future, it will be necessary to investigate whether OX2R in the brain is involved in this interplay.

Animal experiments on acupuncture were performed under restraint or anesthesia, and experiments on stress have not been conducted under physiological conditions. One of the reasons we used PTN in the present study was to verify the antistress effect under a physiological state. After PTN was applied, rats were free to act without restriction. Therefore, a study using PTN is useful for experimentation using acupuncture, as it can be performed without restraint or anesthesia. Despite these advantages, one problem with using PTN is that it should not be applied to an area where the rodents can reach, as they will remove it. Therefore, the types of acupuncture points that can be tested are limited to those on the head and back, as was used this time.

As a limitation, it is unclear how a PTN stimulus was transmitted to the central nervous system from GV20 or how the function of the hypothalamus was altered. In recent years, experiments on analgesic effects have introduced phenomena occurring at acupuncture points. Acupuncture points are places where mast cells form [58,59], and acupuncture stimulation induces mast cell degranulation [60]. For example, adenosine and histamine secreted by mast cells act on their receptors at nerve endings and provide analgesic effects by increasing β-endorphin secretion in the central nervous system [61,62]. In the future, we plan to investigate the involvement of mast cells and the chemical mediators that are secreted from them under the antistress effect of PTN.

## 5. Conclusions

Our findings suggest that PTN treatment at the GV 20 acupuncture point exerts antistress effects via the regulation of orexin secretion. The longer the PTN was, the more effective it was in stimulating antistress effects in the SIS model rat. These findings confirm the effectiveness of PTN and elucidate the role of orexin in controlling the stress response, which may lead to targeted therapies with broad applications in psychiatric health care.

## Figures and Tables

**Figure 1 healthcare-09-00503-f001:**
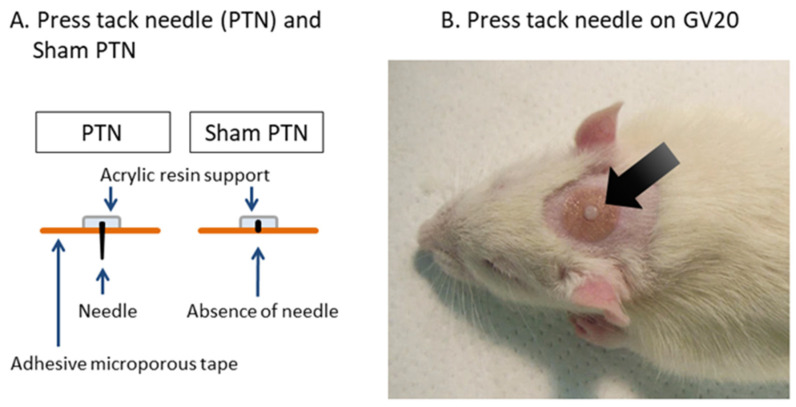
Press tack needle and acupuncture point GV20. (**A**) Schemas of PTN and sham PTN. (**B**) PTN was fixed on the acupuncture point that corresponds to human GV20 (Baihui, black arrow) acupuncture point. GV20, Governor Vessel 20; PTN, press tack needle.

**Figure 2 healthcare-09-00503-f002:**
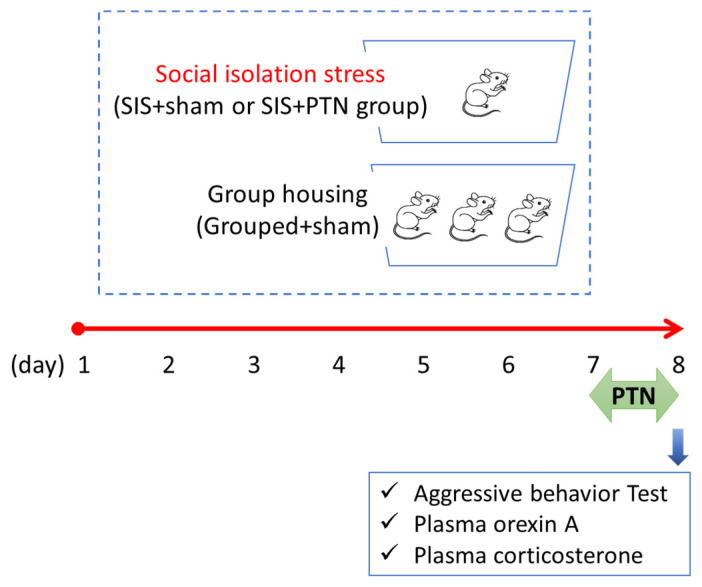
Protocol of the social isolation stress experiment. Rats in the SIS groups were individually housed for eight days. On day 8, the aggressive behavior test was performed, and plasma levels of orexin A and corticosterone were measured. PTN, press tack needle; SIS, social isolation stress.

**Figure 3 healthcare-09-00503-f003:**
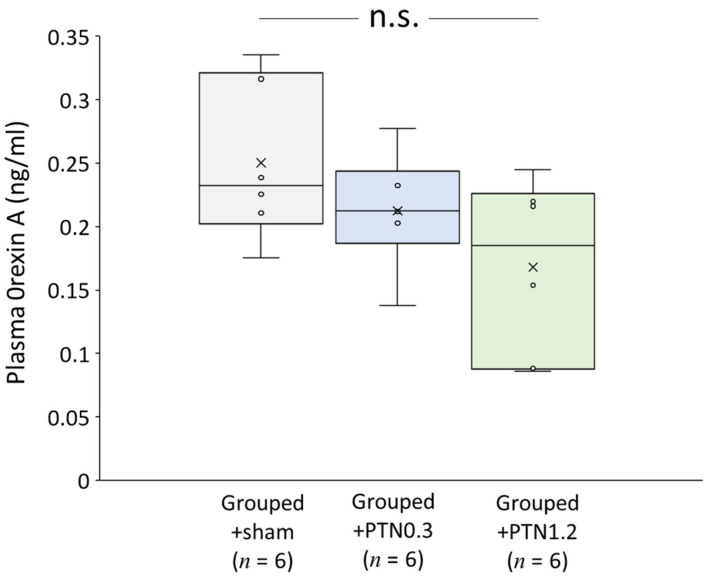
Influence of PTN on the plasma orexin A level. PTNs (lengths: 0.3, and 1.2 mm) were fixed on the acupuncture point GV20. There were no significant differences among the groups. Horizontal lines within boxes denote median values, and x-marks denote the mean values. PTN, press tack needle; n.s., not significant.

**Figure 4 healthcare-09-00503-f004:**
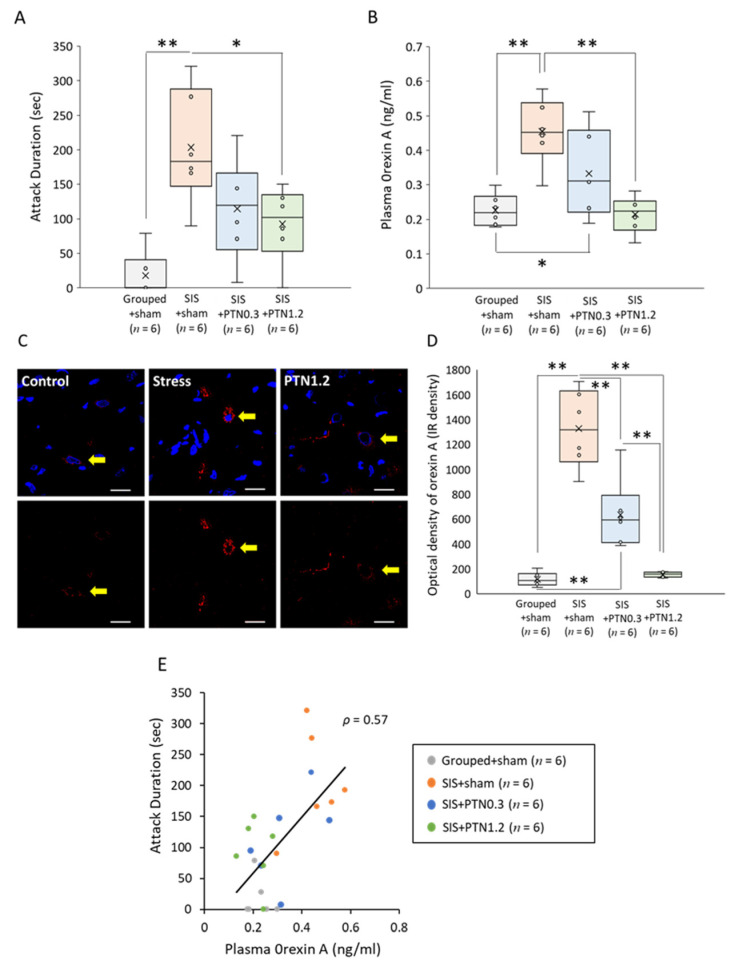
Aggressive behavior test and plasma orexin A level following the social isolation stress load. (**A**) Attack duration during the 10 min aggressive behavior test. Increased attack duration induced by social isolation stress was significantly inhibited after PTN treatment (1.2 mm) (* *p* < 0.05). (**B**) Plasma orexin A level. Increased orexin A level induced by social isolation stress was significantly inhibited due to PTN treatment (1.2 mm) (** *p* < 0.01). Horizontal lines within boxes denote median values, and x-marks denote mean values. (**C**) Immunofluorescent staining of orexin A in in the lateral hypothalamus. The upper row includes orexin A and nuclei, and the lower row includes only orexin A. Red, orexin A; blue, DAPI (nuclei). Yellow arrows indicate orexin neurons, and the white scale bar is 20 μm. (**D**) The fluorescence intensity quantification of orexin A immune reactivity (IR). Increased orexin A level induced by social isolation stress was significantly inhibited due to PTN treatment (** *p* < 0.01). (**E**) Correlation between the attack duration and the orexin A level. A significantly positive correlation was observed between them (*ρ* = 0.57, *p* < 0.01). * *p* < 0.05, ** *p* < 0.01. SIS, social isolation stress; PTN, press tack needle.

**Figure 5 healthcare-09-00503-f005:**
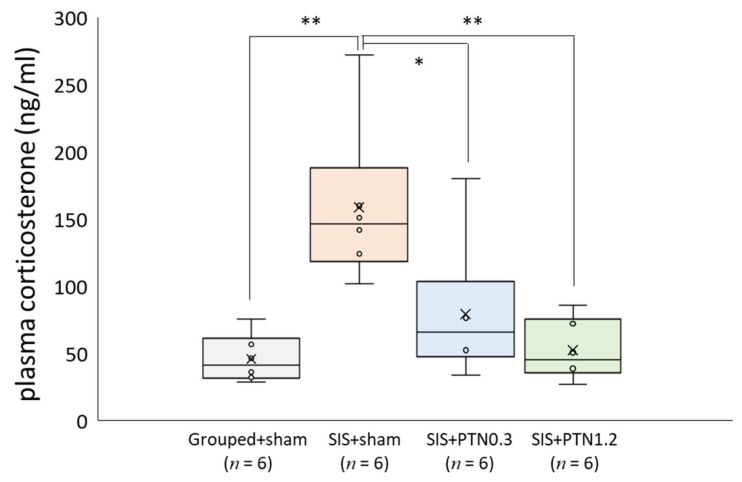
Plasma corticosterone level following the social isolation stress load. Increased corticosterone level induced by social isolation stress was significantly inhibited due to PTN treatment (0.3 mm, * *p* < 0.05; 1.2 mm, ** *p* < 0.01). Horizontal lines within boxes denote median values, and x-marks denote mean values. * *p* < 0.05, ** *p* < 0.01. SIS, social isolation stress; PTN, press tack needle.

**Figure 6 healthcare-09-00503-f006:**
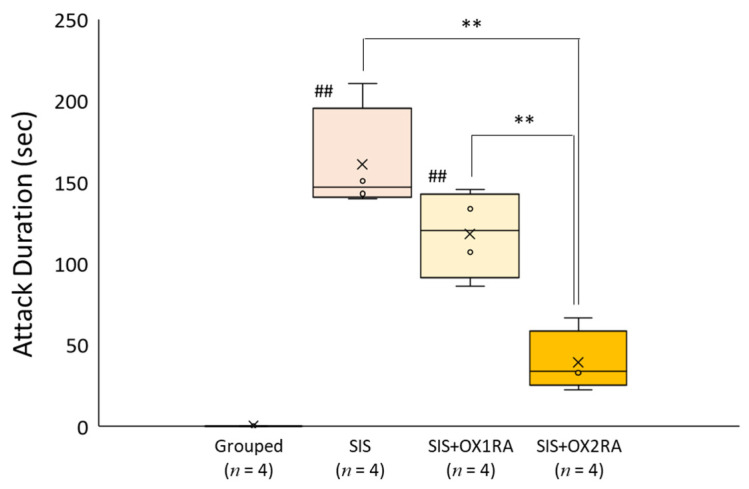
Aggressive behavior test following the treatment of orexin receptor antagonists. Attack durations during the 10 min aggressive behavior test. Increased attack duration induced by social isolation stress was significantly inhibited due to OX2RA treatment (** *p* < 0.01). Horizontal lines within boxes denote median values, and x-marks denote mean values. ** *p* < 0.01, ^##^
*p* < 0.01 vs. Grouped. SIS, social isolation stress; OX1RA, orexin-1 receptor antagonist; OX2RA, orexin-2 receptor antagonist.

**Table 1 healthcare-09-00503-t001:** Acupuncture treatments for rodent stress models.

Authors	Types of Stressor	Ways of Stimulation	Acupuncture Points	Results Effects on the HPA/SAM Axis
Eshkevari et al. [26]	cold temperature	electroacupuncture (10 Hz)	ST36	CRH↓ ACTH↓ CORT↓ NPY↓
Park et al. [27]	maternal separation	twirling	HT7	ACTH↓ CORT↓
Lee et al. [28]	repeated injection of CORT	needle retention	PC6	CRH↓
Yang et al. [29]	forced immobilization	electroacupuncture (3 Hz)	HT3 and PC6	NA↓ A↓
Han et al. [30]	tooth-pulp stimulation	electroacupuncture (3 Hz)	LI4	ACTH↓ CORT↓ NA↓ DA↓
Tanahashi et al. [31]	water-immersion	needle retention	GV20 and Ex-HN3	CORT↓

ST36, stomach 36 (Zusanli); HT7, heart 7 (Shenmen); PC6, pericardium 6 (Neiguan); HT3, heart 3 (Shaohai); LI4, large intestine 14 (Hegu); GV20, Governor Vessel 20 (Baihui); Ex-HN3, extraordinary points of the head and neck 3 (Yintang); HPA axis, hypothalamic-pituitary-adrenal axis; SAM axis, sympathetic-adreno-medullar axis; CRH, corticotropin-releasing hormone; ACTH, adrenocorticotropic hormone; CORT, corticosterone; NPY, neuropeptide Y; NA, noradrenaline; A, adrenaline; DA, dopamine.

## Data Availability

The data presented in this study are available upon request from the corresponding author.

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
