# Peer review of "Regulatory Role of Orexin in the Antistress Effect of “Press Tack Needle” Acupuncture Treatment"

_healthcare, 2021, doi:10.3390/healthcare9050503_

Round 1
Reviewer 1 Report
Authors: Aki Fujiwara et al
Title: Antistress effect of “press tack needle” acupuncture treatment 2 caused by the regulation of orexin secretion
This work aims to investigate the role of orexin and the antistress effect of the press, tack needle acupuncture treatment using rats with social isolation stress.
This work is exciting and has the novelty of finding out more about the orexin system's role in human health, disease, and treatment.
The authors measured stress behavior based on the time the rats showed aggressive behavior and plasma corticosterone and orexin-A levels. Also, the orexin-1 receptor or orexin-2 receptor antagonist was administered to rats that were exposed to stress.
The duration of aggressive behavior was significantly prolonged in the Stress group, and the prolonged duration was inhibited in the PTN (1.2 mm) group. The levels of plasma corticosterone and orexin-A were significantly increased in the Stress group; however, these increases were inhibited in the PTN group. The aggressive behavior was significantly reduced after the orexin-2 receptor antagonist was administered. These findings suggest that PTN treatment at GV20 may have an antistress effect, and the control of orexin is a mechanism underlying this phenomenon.
The manuscript is beautifully written and presented. The methods and experimental designs are excellent.
Comments
The title of the article is not reflecting the true finding of the study.
Suggestion: Regulatory role of Orexin in the antistress effect of “press tack needle” acupuncture treatment
Immunohistochemical analysis revealed increased orexin-A production in the lateral hypothalamus of rats in the Stress group compared with those of the Control group; however, this increase was inhibited in the PTN1.2 group. Representative pictures are shown in Figure 4C. The authors are not given any quantitative data for this. The orexin neuron counting will be done (if complete count not possible, a section match will be enough).
In the result section, there are two same subtitles 3.5. Plasma corticosterone level and 3.6. Plasma corticosterone level. Adding what is referring to will be good to understand for readers.
Line 345, reference 55, check the reference format.
Author Response
We would like to thank you for your review. Since the group names were changed according to the advice of Reviewer 2, all graphs were replaced. We revised our manuscript as described in our responses below.
Point 1: The title of the article is not reflecting the true finding of the study.
Suggestion: Regulatory role of Orexin in the antistress effect of “press tack needle” acupuncture treatment
Response 1: Your recommended title communicates the purpose of our study very well. Per your recommendation, we have corrected it.
Point 2: Immunohistochemical analysis revealed increased orexin-A production in the lateral hypothalamus of rats in the Stress group compared with those of the Control group; however, this increase was inhibited in the PTN1.2 group. Representative pictures are shown in Figure 4C. The authors are not given any quantitative data for this. The orexin neuron counting will be done (if complete count not possible, a section match will be enough).
Response 2: The difference could not be distinguished at a low magnification, but the difference in the secretion from the orexin neurons was confirmed at a high magnification. Therefore, we randomly selected 10 cells per rat and compared the expressions of orexin in individual cells based on the fluorescence intensity.
Point 3: In the result section, there are two same subtitles 3.5. Plasma corticosterone level and 3.6. Plasma corticosterone level. Adding what is referring to will be good to understand for readers.
Response 3: I extend my sincerest apologies for this oversight regarding the numbers for the same subheading. The corrections have been made.
Point 4: Line 345, reference 55, check the reference format.
Response 4: I corrected the format.
Reviewer 2 Report
Main concern:
Possible involvement of orexin system in acupuncture-induced stress relief is interesting and worth examining. Unfortunately, however, this manuscript is too much premature.
1) Result section 3.6 should describe effect of orexin receptor blockers. However, text is an exact copy of section 3.5. I don’t believe that all the authors read and approve the manuscript.
2) DMSO is highly toxic substance. Actually, shorter attack duration (~60 sec) in the stress group in Fig 6 than that in Fig 4 (~200 sec) indicated motor disorder by DMSO. DMSO should be diluted to the concentration that would not affect the observed parameters by itself. Additional experiments are required because this experiment is the only experiment addressing possible causative role of orexin (see also comment #4).
3) Typical photograph (Fig 4C) cannot evidence an increase of orexin by stress and a decrease by PTN. It is definitely required to count fluorescence intensity and/or positive cell numbers together with statistical treatment.
4) Orexin in plasma and LHA was sampled at 2 h after the aggressive behavior test. Therefore, changes in orexin should be considered as the results of aggressive behavior but not the cause of it since the cause is always precede the phenomenon. This weak point should be clearly stated.
5) Results of ANOVA (F, degree of freedom, and p) should be included.
Minor points:
1) Why was PTN 0.6 performed only in the stress-free condition? It may be easy to follow when PTN 0.6 data was deleted.
2) Method for stress-free experiment (Fig 3) is completely missing. Did control rats receive sham PTN? When plasma was sampled?
3) I recommend to use “grouped + sham”, “SIS + sham”, “SIS + PTN0.3”, “SIS + PTN1.2” instead of “control”, “stress”, “PTN0.3”, “PTN1.2” for better contrast and easy understanding.
4) Line 142: The authors wrote “Twenty-four rats were divided into groups of seven”. However, only four groups (control, stress, PTN0.3, and PTN1.2) were described. Do you intended to write “groups of four”?
Author Response
We would like to thank you for your review.
The title of the manuscript has been revised according to the recommendation of Reviewer 1.
Point 1: Result section 3.6 should describe effect of orexin receptor blockers. However, text is an exact copy of section 3.5. I don’t believe that all the authors read and approve the manuscript.
Response 1: Corrections have been made, and all authors have read and approved the revised manuscript. I extended my sincerest apologies for this error.
Point 2: DMSO is highly toxic substance. Actually, shorter attack duration (~60 sec) in the stress group in Fig 6 than that in Fig 4 (~200 sec) indicated motor disorder by DMSO. DMSO should be diluted to the concentration that would not affect the observed parameters by itself. Additional experiments are required because this experiment is the only experiment addressing possible causative role of orexin (see also comment #4).
Response 2: Certainly, when comparing the values in the stress group, your indication is reasonable. TCS OX2 29 is water-soluble, but there was no water-soluble OX1R antagonist. SB334867 dissolves only 5 mg in 1 mL of DMSO, and when SB334867 was administered to the rat at 20 mg / kg, DMSO of 4 mL / kg was required. The dose was determined based on the experiments that were conducted using the same administration method [38,46]. Rats in all groups were given the same dose of DMSO and the results were compared between groups. If this method was inappropriate, or the data is inadequate, we will consider using the dilution method of SB334867 and perform the experiment. If that is required for publication,we request an extension period of one month.
Point 3: Typical photograph (Fig 4C) cannot evidence an increase of orexin by stress and a decrease by PTN. It is definitely required to count fluorescence intensity and/or positive cell numbers together with statistical treatment.
Response 3: The difference could not be distinguished at a low magnification, but the difference in the secretion from the orexin neurons was confirmed at a high magnification. Therefore, we randomly selected 10 cells per rat and compared the expressions of orexin in individual cells based on the fluorescence intensity.
Point 4: Orexin in plasma and LHA was sampled at 2 h after the aggressive behavior test. Therefore, changes in orexin should be considered as the results of aggressive behavior but not the cause of it since the cause is always precede the phenomenon. This weak point should be clearly stated.
Response 4: We previously have measured the orexin levels in the same animal model on a day without aggressive behavior tests. Then, the orexin level significantly increased due to isolated stress. Therefore, the orexin secretion increased regardless of the aggressive behavior test. In other words, we think that the orexin secretion increased due to isolated stress loading. As a result, aggressive behavior was induced, which was proven by the suppression of aggressive behavior after the administration of antagonists prior to the behavior test. We added this to the discussion section (Line 732-).
Point 5: Results of ANOVA (F, degree of freedom, and p) should be included.
Response 5: We have added the results.
Minor points:
Point 1: Why was PTN 0.6 performed only in the stress-free condition? It may be easy to follow when PTN 0.6 data was deleted.
Response 1: In the future, we plan to investigate the effect of PTN0.6 on stress. As pointed out, the PTN0.6 data was deleted.
Point 2: Method for stress-free experiment (Fig 3) is completely missing. Did control rats receive sham PTN? When plasma was sampled?
Response 2: We have mentioned in Line 128 “A PTN of each length was attached to the rats in the PTN group, and a sham PTN was attached to the rats in the Sham group.”
Point 3: I recommend to use “grouped + sham”, “SIS + sham”, “SIS + PTN0.3”, “SIS + PTN1.2” instead of “control”, “stress”, “PTN0.3”, “PTN1.2” for better contrast and easy understanding.
Response 3: We completely agree with you. We revised the group names as suggested.
Point 4: Line 142: The authors wrote “Twenty-four rats were divided into groups of seven”. However, only four groups (control, stress, PTN0.3, and PTN1.2) were described. Do you intended to write “groups of four”?
Response 4: Yes. this should have been written as “groups of four.” This has been corrected in the revised manuscript.
Round 2
Reviewer 2 Report
Although most of my previous concerns were adequately resolved, one most important point has not been cleared. You should use a vehicle that will show no indication of side effect.
For delivery of hydrophobic drug and to reduce dosing of DMSO, you can use 2-hydroxypropyl beta-cyclodextrin (HPbC), a biologically safe compound for such purpose (https://en.wikipedia.org/wiki/Cyclodextrin). Examples of use of HPbC with SB334867 showed successful reduction of DMSO to be 25-30% (Physiol Behav 131:7-16, J Appl Physiol 103:1772-9) without any aversive effect of vehicle.
Author Response
Point 1: For delivery of hydrophobic drug and to reduce dosing of DMSO, you can use 2-hydroxypropyl beta-cyclodextrin (HPbC), a biologically safe compound for such purpose (https://en.wikipedia.org/wiki/Cyclodextrin).
Examples of use of HPbC with SB334867 showed successful reduction of DMSO to be 25-30% (Physiol Behav 131:7-16, J Appl Physiol 103:1772-9) without any aversive effect of vehicle.
Response 1: Thank you for your polite advice. SB334867 was diluted with reference to ‘Physiol Behav 131: 7-16’ (added to the reference). As you say, the SIS group showed aggressive behavior without any aversive effect. Clear results were obtained with (n = 4) each group, and there was a statistically significant difference. The previous data and the current data are shown below. I'm grateful to you.
corrected; L112-115, L208-218, L299-307 and L319

Round 3
Reviewer 2 Report
Congratulations. I'm happy to know that the suggested solvent worked well.
This manuscript is a resubmission of an earlier submission. The following is a list of the peer review reports and author responses from that submission.